# Mindfulness Promotes Online Prosocial Behavior via Cognitive Empathy

**DOI:** 10.3390/ijerph18137017

**Published:** 2021-06-30

**Authors:** Yiqing Lv, Xiuqing Qiao, Jie Leng, Yuanhua Zheng, Qingke Guo

**Affiliations:** 1Department of Psychology, Shandong Normal University, Jinan 250014, China; sdnulvyiqing@163.com (Y.L.); qiaoxiuqing1990@163.com (X.Q.); 17862966905@163.com (J.L.); 17862970224@163.com (Y.Z.); 2Department of Psychology, Guangxi Normal University, Guilin 541004, China

**Keywords:** mindfulness, online prosocial behavior, empathy, mediating effect

## Abstract

Mindfulness plays an important role in promoting prosocial behavior and well-being. With the spread of Internet usage, people’s online prosocial behavior (OPB) has garnered great attention. Based on the link between online and offline behaviors, we predict that mindfulness can also facilitate OPB. We examined the association of mindfulness and OPB and the mediating effect of empathy. A total of 674 Chinese undergraduate students completed self-report measures of these constructs. The results showed that different dimensions of mindfulness predicted empathy, which in turn predicted OPB. Perspective taking was the main mediator in the mindfulness-OPB link. Mindfulness improves receptiveness to others’ needs and feelings, thereby enhancing the willingness to help them, even in none-face-to-face situations.

## 1. Introduction

Mindfulness involves paying attention to the present moment and accepting any thoughts or feelings without judgment [1]. Existing studies have proven that mindfulness-based intervention programs can significantly reduce negative emotions, enhance the ability of emotion regulation, and improve happiness and self-reported quality of life [2,3]. However, few studies have focused on the relationship between mindfulness and online prosocial behavior (OPB). OPB refers to behavior performed voluntarily to help others online without expectation of any reward [4]. Based on the co-construction theory and a negative association between dispositional mindfulness and cyberbullying [5,6,7], and the negative relationship between online prosocial behavior and cyberbullying [8], we predicted that mindfulness can also facilitate online prosociality as it does in offline settings, with empathy playing a mediating role.

### 1.1. Mindfulness and Prosocial Behavior

Mindfulness involves paying attention to the present moment without judgment [9]. The primary goal of mindfulness training focuses on attending and discerning phenomena precisely so that the cause–effect relations are better understood and the characteristics of existence (impermanence, emptiness, and unreliability) are seen clearly [10]. Mindfulness originated from Buddhism, one of the most prosocial religions in the world. Mindfulness can calm the mind and enable people to pay more attention to the needs of others, thereby generating more positive responses aimed at enhancing the well-being of others [11]. This suggests that mindfulness has a great potential to promote prosociality, which has been supported by many empirical studies [12,13,14]. For example, dispositional mindfulness is predictive of other-rated and self-reported prosocial behavior [15]. Mindfulness practitioners are more likely to give up their seats to strangers [16]. Social competence and prosocial outgrowth can also be found in elementary school children who engaged in mindfulness training [17]. Mindfulness interventions, regardless of whether moral emotions were cultivated or not, showed robust effectiveness in promoting prosociality [15].

### 1.2. Empathy as a Mediator

Empathy refers to the capacity to share the emotional state and adopt the perspectives of others [18]. The cognitive component of empathy involves perceiving and reasoning others’ intentions, thoughts, or beliefs, while the emotional component of empathy involves other-oriented emotional responses originating from the perception of another people’s emotional state. Empathy has been considered as the emotional and motivational basis for moral development [19].

Mindfulness training can cultivate empathy. Highly mindful individuals are more responsive and receptive to the needs and feelings of others [20]. Evidence shows that mindfulness training can enhance empathic responses [21,22], and empathy can motivate people to alleviate the pain of others [23]. Therefore, we propose that mindfulness can increase OPB (e.g., online charitable donation, online helping) via enhanced empathic abilities [24].

### 1.3. This Study

Previous studies have shown that mindfulness is positively associated with prosocial behavior in the real world, but whether this association can be extended to the online setting is still not known. It is suggested that social skills (e.g., emotional regulation, empathy with others) developed in offline setting can be applied to the online world [6,25]. Some researchers found that individuals who reported more prosocial behavior (e.g., helping the needy or those in trouble) in real life settings were more likely to help others (e.g., offer help, cheer someone up) in cyberspace [26]. Evidence also shows that mindfulness can enhance emotion regulation by reducing impulsiveness in aggressive environments [27]. Highly mindful individuals tend to have better social relationships and less interpersonal conflicts [28]. Therefore, they are less likely to perpetrate cyberbullying [7]. In addition, people with high levels of mindfulness who were exposed to aggressive online environments showed less retaliatory behavior [27]. Based on this theorizing, we propose that trait mindfulness is predictive of OPB (Hypothesis 1).

In online settings, empathic experiences are also important in establishing strong social ties with others [29]. Online social interactions require understanding others’ feelings, sharing information, and sharing others’ experiences [30]. In other words, OPB can also be motivated by empathy. Thus, we propose that the effect of mindfulness on OPB is mediated by empathy (Hypothesis 2).

In this study, the four dimensions of empathy (empathic concern, perspective taking, fantasy, and personal distress) were analyzed separately because they exert different influences on prosocial behavior [31,32]. A more comprehensive measure developed by factor analyzing other mindfulness measures was used, enabling us to examine the effects of different mindfulness dimensions on OPB [33].

## 2. Materials and Methods

### 2.1. Participants and Procedure

The participants were 674 college students (including liberal arts students from the departments of History and Chinese, and science students from the departments of Physics and Electronic Science) who were to take a group test. None of the students had previously been exposed to mindfulness training or similar practices (e.g., Yoga, Qqigong, Tai Chi). The sample included 394 women and 280 men, with a mean age of 18.5 (SD = 0.74), ranging from 17 to 21 years. The measures were administrated in regular classrooms at the end of an evening study session, with the assistance of two trained research assistants. The participants were told that the measures involved true feelings and experiences in daily life, and were encouraged to complete all items honestly. The administration procedure lasted for approximately 20 min. Each participant was given five gel pens for compensation; all participants participated actively because the prize was tempting. All procedures performed in this study involving human participants were in accordance with the ethical standards of the academic committee at Shandong Normal University and with the 1964 Helsinki declaration and its later amendments or comparable ethical standards.

### 2.2. Measures

#### 2.2.1. Mindfulness

The Five Facets of Mindfulness Questionnaire (FFMQ) consists of 39 items, each one rated using a Likert-type scale ranging from 1 (never or very rarely true) to 5 (very often or always true) [34]. The Chinese version has been proved to have high reliability and validity [35]. The five facets are Observing (noticing or attending to internal and external experiences such as sensations, thoughts, or emotions), Describing (labeling internal experiences with words), Acting with awareness (focusing on one’s activities in the moment as opposed to behaving mechanically), Non-judging of inner experience (taking a non-evaluative stance toward thoughts and feelings), and Non-reactivity to inner experience (allowing thoughts and feelings to come and go, without getting caught up in or carried away by them). In this study, Cronbach’s alpha was 0.82 (Observing), 0.80 (Describing), 0.88 (Acting with awareness), 0.73 (Non-judging), and 0.72 (non-reactivity).

#### 2.2.2. Empathy

The Interpersonal Reactivity Index (IRI) is a 5-point-Likert-type scale (1 = strongly disagree, 5 = strongly agree), designed to measure dispositional empathy [31]. The Chinese version of the IRI consists of 22 items, and it shows acceptable reliability and validity in college students [36]. It includes four distinct components of empathy: Empathic Concern (feeling emotional concern for others), Perspective Taking (cognitively taking the perspective of another), Personal Distress (negative feelings in response to the distress of others), and Fantasy (emotional identification with characters in books, films, etc.). In this study, the Cronbach’s alpha of Empathic Concern, Perspective Taking, Personal Distress, and Fantasy were 0.66, 0.80, 0.81 and 0.67, respectively.

#### 2.2.3. OPB

The Internet Altruistic Behavior Scale (IABS), a measure developed in China [37], contains 26 4-point (1 = never, 4 = always) items. In this study, 14 items of the IABS were adopted, measuring three dimensions of OPB (i.e., online support, online guidance, and online sharing). The online support dimension contains 6 items (e.g., Caring and encouraging others). The online guidance dimension contains 4 items (e.g., Guiding others to use the Internet more efficiently). The online sharing dimension contains 4 items (e.g., sharing with others experiences of successful learning). In this study, Cronbach’s alpha of the whole scale was 0.89.

## 3. Results

### 3.1. Descriptive Statistics and Correlation Analysis

Means, standard deviations, and Pearson correlations among variables are presented in Table 1. As can be seen from the table, online prosocial behavior not only has a significant positive correlation with all five dimensions of mindfulness, but also has a significant positive correlation with the four dimensions of empathy.

### 3.2. Mediation Analysis Using Empathy as Mediators

AMOS (version 21.0) (IBM, Armonk, NY, USA) was used to conduct mediation analysis regarding the role of empathy in the mindfulness-OPB association. The mediating effect was tested by the Bootstrap method with 5000 repeated samples [38].

#### 3.2.1. Observing

After controlling for the effects of gender, age, and monthly household income, the hypothesized model (Figure 1) fitted the data well (χ^2^/df = 4.55, CFI = 0.92, RMSEA = 0.07). The effect of gender was significant (β_1_ = −3.38, *p* < 0.05). The effect of age (β_2_ = 0.34, *p* > 0.05) and monthly household income (β_3_ = −0.15, *p* > 0.05) were non-significant. The total (β = 0.37, SE = 0.04, *p* < 0.05, 95% CI = [0.29, 0.43]), direct (β = 0.22, SE = 0.04, *p* < 0.05, 95% CI = [0.15, 0.30]), and total indirect (β = 0.14, SE = 0.03, *p* < 0.05, 95% CI = [0.09, 0.19]) effect of Observing on OPB were all significant. The mediating role of perspective taking was statistically significant.

#### 3.2.2. Describing

After controlling for the effects of gender, age and the monthly household income, the hypothesized model (Figure 2) fitted the data well (χ^2^/df = 5.06, CFI = 0.97, RMSEA = 0.07). The effect of gender was significant (β_1_ = −3.46, *p* < 0.05). The effect of age (β_2_ = 0.40, *p* > 0.05) and monthly household income (β_3_ = −0.09, *p* > 0.05) were non-significant. The total (β = 0.35, SE = 0.04, *p* < 0.05, 95% CI = [0.28, 0.42]), direct (β = 0.25, SE = 0.04, *p* < 0.05, 95% CI = [0.18, 0.32]), and total indirect (β = 0.10, SE = 0.02, *p* < 0.05, 95% CI = [0.06, 0.15]) effect of Describing on OPB were all significant. The mediating role of perspective taking and fantasy were statistically significant.

#### 3.2.3. Acting with Awareness

After controlling for the effects of gender, age, and monthly household income, the hypothesized model (Figure 3) fitted the data well (χ^2^/df = 4.12, CFI = 0.92, RMSEA = 0.07). The effect of gender was significant (β_1_ = −3.41, *p* < 0.05). The effect of age (β_2_ = 0.37, *p* > 0.05) and monthly household income (β_3_ = −0.11, *p* > 0.05) were non-significant. The total (β = 0.15, SE = 0.04, *p* < 0.05, 95% CI = [0.07, 0.23]), direct (β = 0.10, SE = 0.04, *p* < 0.05, 95% CI = [0.03, 0.17]), and total indirect effect (β = 0.06, SE = 0.02, *p* < *0*.05, 95% CI = [0.01, 0.10]) of Acting with awareness on OPB were all significant. The mediating role of perspective taking was statistically significant.

#### 3.2.4. Non-Judging

After controlling for the effects of gender, age, and monthly household income, the hypothesized model (Figure 4) fitted the data well (χ^2^/df = 4.94, CFI = 0.89, RMSEA = 0.07). The effect of gender was significant (β_1_ = −3.21, *p* < 0.05). The effect of age (β_2_ = 0.31, *p* > 0.05) and monthly household income (β_3_ = −0.12, *p* > 0.05) were non-significant. The total (β = 0.10, SE = 0.05, *p* < 0.05, 95% CI = [0.01, 0.20]) and direct (β = 0.08, SE = 0.04, *p* < 0.05, 95% CI = [0.01, 0.15]) effect of Non-judging on OPB were significant. The total indirect effect was non-significant (β = 0.03, SE = 0.02, *p* > 0.05, 95% CI = [−0.02, 0.08]).

#### 3.2.5. Non-Reactivity to Inner Experience

After controlling for the effects of gender, age, and monthly household income, the hypothesized model (Figure 5) fitted the data well (χ^2^/df = 4.42, CFI = 0.91, RMSEA = 0.07). The effect of gender was significant (β_1_ = −3.23, *p* < 0.05). The effect of age (β_2_ = 0.24, *p* > 0.05) and monthly household (β_3_ = −0.14, *p* > 0.05) income were non-significant. The total (β = 0.22, SE = 0.04, *p* < 0.05, 95% CI = [0.14, 0.31]), direct effect (β = 0.10, SE = 0.04, *p* < 0.05, 95% CI = [0.02, 0.18]), and total indirect effect (β = 0.13, SE = 0.02, *p* < 0.05, 95% CI = [0.08, 0.18]) of Non-reactivity to inner experience on OPB were all significant. The mediating role of perspective taking and fantasy were statistically significant.

## 4. Discussion

This study found that all dimensions of trait mindfulness measured by the FFMQ predicted more OPB (supporting Hypothesis 1). This suggests that individuals who notice a wide range of social stimuli are more likely to act prosocially, even when the recipients of help are remote others [15]. This study provides more evidence that online and offline prosocial behaviors are connected [6].

We found that different dimensions of the FFMQ (except for the non-judging of mindfulness) influenced OPB via empathy (supporting Hypothesis 2). This is consistent with previous findings that mindfulness facilitates prosocial responses to socially ostracized strangers via empathy concern, suggesting that clear awareness of one’s own feelings facilitates empathic responses to others’ pain [20,22,39]. High mindfulness individuals can observe and describe accurately inner and outer experiences in the present moment, and allow the thoughts and feelings to come and go freely. Therefore, they are more responsive and receptive to the needs and feelings of others [20]. However, Non-judging had only a small influence on OPB, which was not mediated by empathy. Letting experiences pass by like a cloud without judging whether they are positive or negative and right or wrong may be unbeneficial [34]. This may mean attending to other people’s pain indifferently.

Notably, two cognitive (perspective taking and fantasy) but not affective (empathic concern and personal distress) components of empathy mediated the mindfulness-OPB association [31]. High mindfulness individuals are better in understanding others’ feelings and thoughts, and they tend to be more helpful [39]. The anonymity and invisibility of cyberspace provides more opportunities for people to communicate and self-disclose, which makes people more willing to express their feelings in words online [29]. Online social interactions do not involve non-verbal information (e.g., facial expressions, tones, body language). People can only understand others through verbal information expressed online [40]. This reduces the roles of empathic concern and personal distress in making helping decisions. Consistent with the results of this study, it has been found that mindfulness has a positive impact on the engagement of nursing professionals, which is more strongly mediated by cognitive empathy compared with affective empathy [41]. This is because mindfulness reduces emotional responses to internal experiences [42].

The limitations of this study must be addressed. A cross-sectional design has limited power to reveal causal links [43], and the inference of effects can be substantially biased because only one-wave data was involved [44]. An experimental paradigm is welcomed. We invite longitudinal studies to investigate the mediation processes that may unfold over time (e.g., mindfulness training promotes empathy abilities and consequently leads to increased OPB). Furthermore, demographically diverse samples from different cultural backgrounds should be involved to make the findings more generalizable.

Widespread use of the Internet raises concern about the incidence of negative online behaviors (e.g., cyberbullying, online crime, internet addiction). The beneficial effects of mindfulness bring about enlightenment on the solution of these problems. Mindfulness training enhances empathic responses to remote others, thereby increasing positive social behaviors and reducing negative social behaviors.

## 5. Conclusions

This study found that a highly mindful individual is more understanding of others’ thoughts and is more likely to take others’ perspectives, thereby tending to lend a helping hand, even when the recipients are remote others. This study has made progress by revealing that mindfulness facilitates prosocial behavior towards remote others, and only perspective taking mediates this relationship. This suggests that cultivating mindfulness and cognitive empathy can enhance online prosociality. We invite future research to further explore other possible mediating mechanisms. In addition, we propose that psychosocial interventions aimed at fostering mindfulness and empathy can increase prosocial behavior in both online and offline settings.

## Figures and Tables

**Figure 1 ijerph-18-07017-f001:**
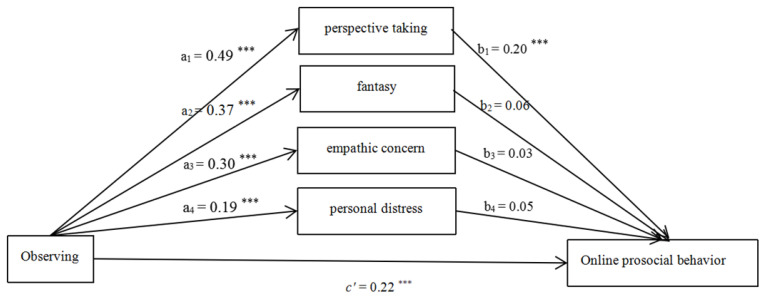
Mediating roles of empathy components (Observing as the predictor). Note: *** *p* < 0.001.

**Figure 2 ijerph-18-07017-f002:**
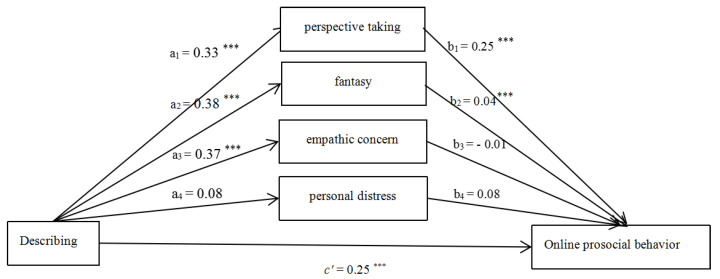
Mediating roles of empathy components (Describing as the predictor). Note: *** *p* < 0.001.

**Figure 3 ijerph-18-07017-f003:**
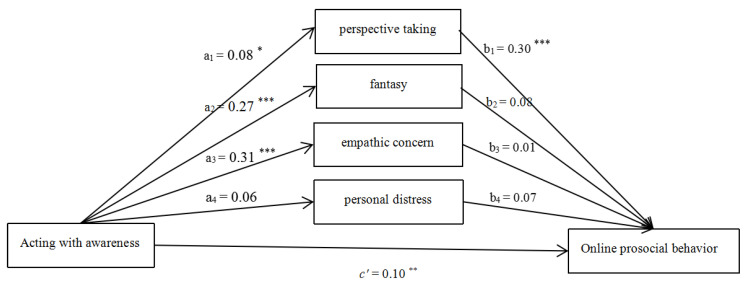
Mediating roles of empathy components (Acting with awareness as the predictor). Note: * *p* < 0.05, ** *p* < 0.01, *** *p* < 0.001.

**Figure 4 ijerph-18-07017-f004:**
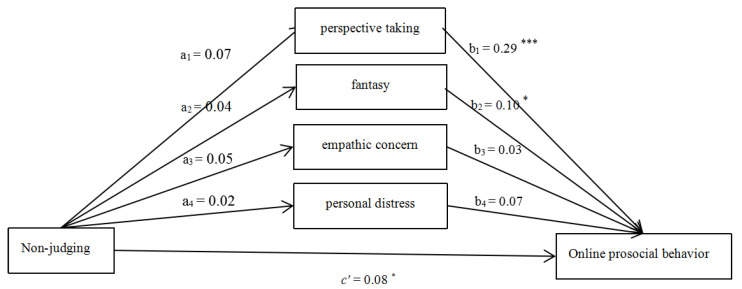
Mediating roles of empathy components (Non-judging as the predictor). Note: * *p* < 0.05, *** *p* < 0.001.

**Figure 5 ijerph-18-07017-f005:**
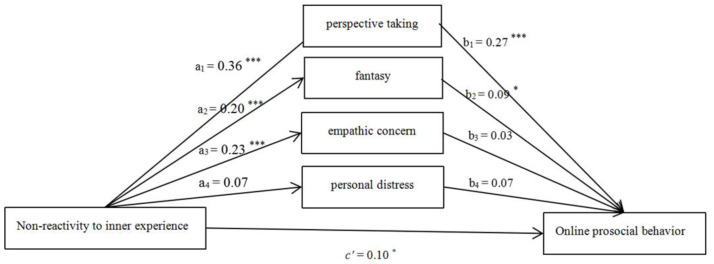
Mediating roles of empathy components (Non-reactivity as the predictor). Note: * *p* < 0.05, *** *p* < 0.001.

**Table 1 ijerph-18-07017-t001:** Means, standard deviations, and Pearson correlations among research variables (*n* = 674).

	1	2	3	4	5	6	7	8	9	10
1 OPB	1									
2 Perspective Taking	0.35 **	1								
3 Empathic Concern	0.18 **	0.34 **	1							
4 Personal Distress	0.12 **	0.07	0.23 **	1						
5 Fantasy	0.22 **	0.35 **	0.59 **	0.39 **	1					
6 Observing	0.36 **	0.49 **	0.30 **	0.19 **	0.38 **	1				
7 Describing	0.34 **	0.33 **	0.37 **	0.08 *	0.38 **	0.43 **	1			
8 Acting with awareness	0.14 **	0.09 *	0.31 **	0.06	0.26 **	0.05	0.47 **	1		
9 Non-judging	0.13 **	0.07	0.1	0.02	0.04	0.01	0.24 **	0.37 **	1	
10.Non-reactivity to innerexperience	0.24 **	0.36 **	0.23 **	0.07	0.20 **	0.49 **	0.34 **	0.10 *	0.11 **	1
M	36.80	17.81	20.96	14.63	20.70	26.71	25.39	24.03	23.38	21.90
SD	9.00	3.73	4.06	4.26	4.41	5.46	5.11	6.13	4.80	4.09

Note: OPB = Online Prosocial Behavior; ** p* < 0.01, *** p* < 0.01.

## Data Availability

Data and research materials are available on our OSF project page: https://osf.io/xuemh/, accessed on 9 April 2021.

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
