# Peer review of "Mindfulness Promotes Online Prosocial Behavior via Cognitive Empathy"

_ijerph, 2021, doi:10.3390/ijerph18137017_

Round 1
Reviewer 1 Report
The matter of the manuscript is interesting. The authors propose that trait mindfulness is predictive of Online Prosocial Behavior (OPB) and that the effect of mindfulness on OPB is mediated by empathy.
The paper is well written and use a good statistical analysys with AMOS. However I have some questions.
Did the authors check the opposite directions of their hypothesis. I mean, Is there a bidirectional relationship between OPB and mindfulness? We know that mindfulness can be trained. Did the authors review the conditions of the participants, in terms of training mindfulness or not? Did the authors check if the participants made some kind of oriental sport, like yoga or similar? These contextual variables could affect to the participants selected.
Others specifics questions:
Line 22: cites (1) and (2) belong to the years 2003 and 2007. Try to look for more recent studies.
Line 77. It is important to know what kind of studies the students developed. Are they from different degrees or similar? Are computer experts or psychologist? This information is relevant according to external validity. More information is need about the selection process. How did the authors recruit them? Did you send an on line questionnaire? Could the participants develop a desirable vias in terms of self-perception?
Line 284 “High mindfulness individuals are better in understanding others' feelings and thoughts, and tend to be helpful[33].” Does the author consider that people who are more proactive are more likely to develop mindfulness training? I mean, could be that people who are proactive would be mor open mind to practice mindfulness. I think it is relevant to know more about the characteristics of the participants in order to control contextual variables that cold affect the mindfulness responses.
Line 108 maybe the Cronbach’s alpha minor than 0.7 could be a possible question to develop in limitations sections: “of Empathic Concern, Perspective Taking, Personal Distress, and Fantasy were .66, .80, .81, and.67, respectively”.
Finally, I think this paper is interesting for the develop of new scientific perspectives on Mindfulness and Prosocial Behavior.
Author Response
The matter of the manuscript is interesting. The authors propose that trait mindfulness is predictive of Online Prosocial Behavior (OPB) and that the effect of mindfulness on OPB is mediated by empathy.
Authors’ responses: Thanks for your comments.
The paper is well written and use a good statistical analysys with AMOS. However I have some questions.
Authors’ responses: Thanks for your comments.
Did the authors check the opposite directions of their hypothesis. I mean, Is there a bidirectional relationship between OPB and mindfulness? We know that mindfulness can be trained. Did the authors review the conditions of the participants, in terms of training mindfulness or not? Did the authors check if the participants made some kind of oriental sport, like yoga or similar? These contextual variables could affect to the participants selected.
Authors’ responses: Thanks for your valuable suggestions. Among the variables studied in this paper, mindfulness can be referred to as a trait, while online prosocial behavior (OPB) is behavior. Logically, in general, most researchers assume that traits influence behavior, but not vice versa. For example, an experimental study has found that participants who received mindfulness training reported more prosocial behavior five days later (Hafenbrack et al., 2020), and studies have found that trait mindfulness or state mindfulness is a strong predictor of prosocial behavior (Berry et al., 2018). Actually there is no evidence that prosocial behavior affects mindfulness. Given that OPB is a form of prosocial behavior, we proposed a one-directional causal relationship between mindfulness and online prosocial behavior.
We responded to the question asking details of participants characteristics and control variables in follow-up paragraphs. Moreover, we made changes in the main text, which can be seen in the Methods section of the manuscript on Page 2-3, Line 87-101.
References
Hafenbrack, A. C.; Cameron, L. D.; Spreitzer, G. M.; Zhang, C.; Noval, L. J.; Shaffakat, S. Helping people by being in the present: Mindfulness increases prosocial behavior. Organizational Behavior and Human Decision Processes2020, 159, 21-38.
Berry, D. R.; Cairo, A. H.; Goodman, R. J.; Quaglia, J. T.; Green, J. D.; Brown, K. W. Mindfulness increases prosocial responses toward ostracized strangers through empathic concern. Journal of Experimental Psychology: General2018, 147, 93.
Others specifics questions:
Line 22: cites (1) and (2) belong to the years 2003 and 2007. Try to look for more recent studies.
Authors’ responses: Thanks for your valuable suggestions. In a clinical context, mindfulness was first conceptualized by Kabat-Zinn as “paying attention in a particular way, on purpose, in the present moment, and non-judgmentally” (Kabat-Zinn 1994). Since then, many definitions and concepts of mindfulness have evolved, and until now, there is still no consensus on it (Baer et al. 2008; Brown et al. 2007). To keep the definition up-to-date, we replaced the 2003 concept with a summary of the concept of mindfulness from a 2019 article. In addition, we also selected the recent literature to replace the 2007 literature.
Specific revisions can be found on Page 1, Line 20-24, and Page 8, Line 272-275.
References
Kabat-Zinn, J. Mindfulness meditation for everyday life. London: Piatkus Books1994.
Baer, R. A.; Smith, G. T.; Lykins, E.; Button, D.; Krietemeyer, J.; Sauer, S.; ... Williams, J. M. G. Construct validity of the five facet mindfulness questionnaire in meditating and nonmeditating samples. Assessment2008, 15, 329-342.
Brown, K. W.; Ryan, R. M.; Creswell, J. D. Mindfulness: Theoretical foundations and evidence for its salutary effects. Psychological Inquiry2007, 18, 211-237.
Line 77. It is important to know what kind of studies the students developed. Are they from different degrees or similar? Are computer experts or psychologist? This information is relevant according to external validity. More information is need about the selection process. How did the authors recruit them? Did you send an on line questionnaire? Could the participants develop a desirable vias in terms of self-perception?
Authors responses: Thanks for your valuable suggestions. We recruited students offline and conducted group questionnaire administration in regular classrooms. Response bias is well controlled because of anonymity of this study. The participants were told that their responses will be kept confidentially. Due to a word limit of this journal, we did not write it in details before. In the revised manuscript, more information about the selection of participants and the measurement process was added on Page 2-3, Line 87-101.
Line 284 “High mindfulness individuals are better in understanding others' feelings and thoughts, and tend to be helpful[33].” Does the author consider that people who are more proactive are more likely to develop mindfulness training? I mean, could be that people who are proactive would be more open mind to practice mindfulness. I think it is relevant to know more about the characteristics of the participants in order to control contextual variables that cold affect the mindfulness responses.
Authors responses: Thanks for your valuable suggestions. The followings are our considerations regarding this issue.
First of all, according to the logic that proactive people are more likely to improve their level of mindfulness and thus are more likely to engage in OPB behavior, proactive personality is likely to be the antecedent variable of mindfulness. However, we have not yet found any empirical studies to demonstrate the significant impact of proactive personality on mindfulness. In addition, we found a low correlation between proactive personality and mindfulness (Bajaba et al., 2018). Therefore, whether proactive personality is the antecedent variable of mindfulness is not clear.
Second, given the large number of participants and the way we collected data (the questionnaires are from several whole class), the proactive personality variable may have balanced effects on independent and dependent variables.
References
Bajaba, S.; Fuller, B.; Marler, L.; Bajaba, A. Does mindfulness enhance the beneficial outcomes that accrue to employees with proactive personalities?. Current Psychology2018, 1-10.
Line 108 maybe the Cronbach’s alpha minor than 0.7 could be a possible question to develop in limitations sections: “of Empathic Concern, Perspective Taking, Personal Distress, and Fantasy were .66, .80, .81, and.67, respectively”.
Authors responses: Thanks for your valuable suggestions. Devellis (2016) suggested the ranges of Cronbach’s alpha for research scales are as follows: “below .60, unacceptable; between .60 and .65, undesirable; between .65 and .70, minimally acceptable; between .70 and .80, respectable; between .80 and .90, very good” . And Boateng et al. (2018) supported using this standard to judge the reliability of scales in health, social, and behavioral research. Therefore, the reliability values (.66 and .67) of the scale in this study are within the acceptable range. However, we are very grateful for the reviewer's reminder. We will pay attention to the reliability of this scale in the subsequent studies.
References
Boateng, G. O.; Neilands, T. B.; Frongillo, E. A.; Melgar-Quiñonez, H. R.; Young, S. L. Best practices for developing and validating scales for health, social, and behavioral research: a primer. Frontiers in public health2018, 6, 149.
DeVellis, R. F. Scale development: Theory and applications (Vol. 26). Sage publications2016. p136.
Finally, I think this paper is interesting for the develop of new scientific perspectives on Mindfulness and Prosocial Behavior.
Authors responses: Thanks for your comments.

Reviewer 2 Report
Dear authors,The title caught my attention. In addition, mindfulness is currently a widely used technique in health. Congratulations for the work done, the methodology used and especially for the novelty that the study presents, which gives us something to reflect on. My suggestions for improvement focus on the introduction and conclusion. In the introduction: In the first paragraph I would like you to be able to explain what are the physical and mental benefits of mindfulness. Then explain what prosocial behavior online is and what cyberbullying has to do with it. In the second paragraph I better explain what mindfulness does in relation to prosocial behavior, I can't quite understand the relationships. Better delimit in your theoretical framework, empathy and prosocial behavior. In conclusion: Add some paragraphs to include: -Theoretical implications, what this means for the academic / scientific world at a theoretical level. -Practical implications, what this means for society. -Future prospects or future lines of research.
Congratulations on the job
Author Response
The title caught my attention. In addition, mindfulness is currently a widely used technique in health. Congratulations for the work done, the methodology used and especially for the novelty that the study presents, which gives us something to reflect on.
Authors responses: Thanks for your comments.
My suggestions for improvement focus on the introduction and conclusion. In the introduction:
In the first paragraph I would like you to be able to explain what are the physical and mental benefits of mindfulness. Then explain what prosocial behavior online is and what cyberbullying has to do with it.
Authors responses: Thanks for your valuable suggestions. According to your suggestion, we have made detailed modifications in the first paragraph (see the blue text in the manuscript), which introduces the definition of online prosocial behavior and its relationship with cyberbullying behavior, and the mindfulness's benefits, respectively. Due to word limit of this journal, we did not give a detailed explanations earlier.
Specific revisions can be found on Page1, Line 20-30.
In the second paragraph I better explain what mindfulness does in relation to prosocial behavior, I can't quite understand the relationships. Better delimit in your theoretical framework, empathy and prosocial behavior.
Authors responses: Thanks for your valuable suggestions. We have accepted your suggestions and made revisions in the second to fourth paragraphs of the main text. It mainly introduces the logical relationship between mindfulness and prosocial behavior, and elaborates the relationship between empathy and prosocial behavior. However, due to the 3000 words limit, we can only summarize it.
Specific revisions can be found on Page 1, Line 37-39; Page 2, Line 54-59.
In conclusion: Add some paragraphs to include: -Theoretical implications, what this means for the academic / scientific world at a theoretical level. -Practical implications, what this means for society. -Future prospects or future lines of research.
Authors responses: Thanks for your valuable suggestions. We have made modifications regarding the theoretical significance, practical significance, and future directions of this study, respectively.
Specific revisions can be found on Page 7, Line 246-254.
